# Digital Health Sensing for Personalized Dermatology

**DOI:** 10.3390/s19153426

**Published:** 2019-08-05

**Authors:** Pier Spinazze, Alex Bottle, Josip Car

**Affiliations:** 1Global Digital Health Unit, Department of Primary Care and Public Health, School of Public Health, Imperial College London, St Dunstan’s Road, London W6 8RP, UK; 2Centre for Population Health Sciences (CePHaS), Lee Kong Chian School of Medicine, Nanyang Technological University, 11 Mandalay Road, Singapore 308232, Singapore

**Keywords:** mobile dermatology, artificial intelligence, digital phenotyping, eczema, atopic dermatitis, digital sensing

## Abstract

The rapid evolution of technology, sensors and personal digital devices offers an opportunity to acquire health related data seamlessly, unobtrusively and in real time. In this opinion piece, we discuss the relevance and opportunities for using digital sensing in dermatology, taking eczema as an exemplar.

Digital health sensing generates a novel category of health data via continuous, unobtrusive detection, collection and measurement of a person’s activity, events and interactions with the environment through digital devices including smartphones, wearables or Internet of Things (IoT). To date, exploration of the potential benefits of this data has mainly been in the mental health area [1]. However, these benefits could extend to other specialties, including personalized dermatology. Taking eczema as an exemplar of a skin condition, we explore how its management could be improved by digital health sensing.

Dermatology requires doctors to make treatment decisions based on patient self-reporting, which poses challenges including patient recall or recognition of exacerbating factors, leading to a trial-and-error approach to management and additional consultations [2].

A chronic skin condition like eczema poses a significant burden on both healthcare resources and patients’ quality of life [3]. Itching is a major symptom of eczema reported in 85% of patients. Quality of life is largely impacted by sleep deprivation due to itching and there is an increased incidence of depression and suicidal thoughts [3]. Itching in eczema follows a cyclical course of flares and periods of remission. Therefore, being able to accurately quantify the amount of scratching over time can aid in developing effective management regimens and alerting patients to anticipate and potentially prevent flare ups [4]. Few objective methods have been developed to quantify and measure patient scratching. Polysomnography, strain gauges, pressure sensitive meters, infrared video, and manual labelling have previously been used to attempt measurement. Some are more accurate than others, however, they have all proved too costly or impractical for routine care [4]. In contrast to traditional methods, new IoT sensors and wearables are lightweight, inexpensive and wireless. There are potential disadvantages and considerations when using wearables including subsequent hyper-allergic responses. Watch strap irritation has been well documented and there are multiple reports in non-peer reviewed literature, including social media and blogs, on wearables causing an irritant contact dermatitis secondary to sweat and friction, or an allergic response to either constituent metals or synthetic rubber parts [5].

Since the early 2000s, several researchers have looked at using wrist-worn actigraphy devices for the detection of scratching. However, as these devices reflect movement, it proved difficult to differentiate between different forms of movement e.g. restless sleep, walking, etc. Recent advances in machine learning algorithms are now able to differentiate and identify these with increasing sensitivity and specificity [6]. The precision and number of personal digital sensors not only allow for non-invasive collection of activity markers, including itching and sleep, but also evaluation of their severity and aggravating factors such as temperature, medications, locations, etc., and they provide real-time insights into responses to treatment. A recent study reported that these sensors had an almost 90% accuracy for detecting scratching [6], which may improve over time due to more data points. This combined with, for example, microphone, temperature, air quality and other sensor data will give a new dimension to the precision of assessment of scratching and other symptoms. Most fitness wearables also couple actigraphy with heart rate measurement. As eczema is a stress-responsive disorder that involves autonomic nervous system dysfunction, heart rate variability could also indicate itching in individuals with eczema. Subjects exhibit an overactive sympathetic response to itching and scratching, while the parasympathetic tone is persistently and rigidly elevated, showing a lack of adaptability in response to stress.

Sleep disturbance, a common consequence of scratching and often an indicator of eczema severity, could also be measured through actigraphy. A lack of sleep, is known to have negative effects on mental, physiological, emotional and social well-being. Beyond actigraphy and wrist worn devices, studies and several commercially available applications have also experimented with utilizing smartphone data to estimate sleep patterns, with varying degrees of success. These leverage off smartphone sensor data including a combination of device “lock” duration, activity via accelerometers (stationary time), microphone (ambient sounds), and camera (ambient light) to approximate the amount of sleep [7]. The benefit of using a person’s phone is the ability to measure without the need or expense of a separate device. However, they rely on the user having the phone next to them during sleep. Advances in IoT further expands on the availability of sensors and provides another means of monitoring. Sensors are being incorporated into everything from appliances to clothing to diapers. Major consumer diaper companies are now venturing into smart diapers to quantify not only babies sleep but also their bowel movements and vital signs [8]. These consumer devices including baby monitors with sound and movement detection, could potentially also be used to detect both itching and sleep. However, regardless of the device or methods used to acquire patient data, this approach to assessment requires validation against a gold standard or validated patient scoring metric. Although there is no globally accepted standardized diagnostic criteria for eczema [9], there are a number of validated instruments for the assessment of severity including the Eczema Area and Severity Index [10] and the Severity Scoring of Atopic Dermatitis Index (SCORAD) [11] which could be used (see Figure 1).

Quantified itching and sleep data could be further enriched and correlated with potential risk factors, thus allowing for evaluation and prediction of causative irritants. Modifying environmental risk factors or exposure plays a substantial role in managing the disease. Environmental factors could be acquired passively, e.g., time of year/season, temperature, humidity, pollen and air pollution as well as location recorded through smartphones, either via sensors, e.g., GPS, Wi-Fi, barometer, etc., or online real-time sources, e.g. weather station reports. Other risk factors could be acquired actively through patient questionnaires. Diaries may be both text-based and visual, i.e., taking images of lesions, meals, medications, etc. In the short term, these images could serve as a visual reminder or reference for the user to evaluate against their objective score. However, with advances in image recognition, future applications may allow for the automatic evaluation of content.

There are important considerations regarding the quality of all this data. Data variability has two sources, including differences in hardware and software (e.g., lack of standardization of sensors or signal processing algorithms) and variability in human behaviour. Sensors in smartphones can vary between manufacturers, models and versions and according to participants’ preferences and behaviours. How people carry their smartphone (pocket, handbag or backpack) can profoundly impact the sensor data, for example, accelerometer movement recordings. Thus, this data needs to be captured in raw format and individualized. Another consideration is user compliance or retention. To capture high quality longitudinal data, the user will need to continuously engage with their device or application.

A number of open source platforms have been developed to do this, including Beiwe [12], LAMP (Learn Assess Manage Prevent) [13] and RADAR-base (Remote Assessment of Disease And Relapses) [14]. Although these platforms were primarily developed for research purposes with mental health in mind, they all perform the same function of amalgamating passively collected sensor data with actively acquired user data from questionnaires and tests to provide a comprehensive picture of an individual’s state of health. These platforms could be leveraged for any condition including eczema. The benefit of these insights is evident to practitioners or researchers, however from a patient’s perspective the value is not as translatable or easily understood. In order to ensure compliance and user participation, these platforms need to provide feedback or engage users through insights, educational outputs or interventions that encourage retention.

The capacity to measure, capture and interpret multiple sources of data using personal devices and sensors opens up new opportunities for preventative health management. To date, this has been explored within the realm of mental health. However, the long-term nature and frequent need for ongoing monitoring of skin conditions provide an opportunity to develop personalized, patient-centered care delivered through digital devices, assisted by artificial intelligence.

## Figures and Tables

**Figure 1 sensors-19-03426-f001:**
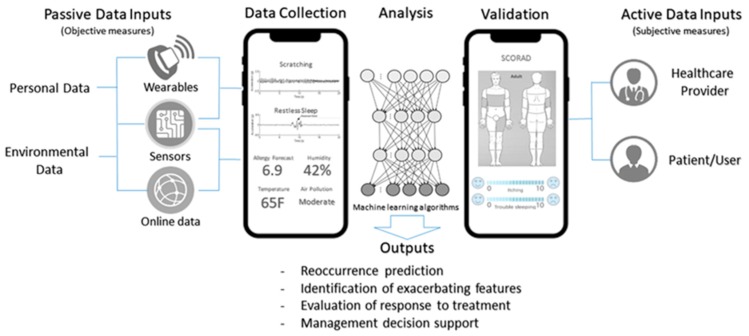
Example of integrated data collection and evaluation using a smartphone.

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
