# Peer review of "Digital Health Sensing for Personalized Dermatology"

_sensors, 2019, doi:10.3390/s19153426_

Round 1
Reviewer 1 Report
The opinion paper proposes to use remote data collection platform for skin diseases. As the idea is sound, authors should give example of existing platforms for data collection considering their pros and cons. e.g.
RADAR-base: An open source platform for collecting, monitoring and analysing using sensors, wearables and mobile devices.
Authors should give a brief insight into the sensor data quality and user compliance in case of skin diseases.
Author Response
Dear Reviewer,
Thank you for your valued comments. We have attached a file documenting all comments addressed in the manuscript.
Kind regards
Pier

Reviewer 2 Report
The authors propose a specific topic of research to enhance the dermathologycal monitoring using sensors and IoT. The idea is well explained and the challenges clear. It is ok for publication.
Some more references wrt wearable solutions for similar problems can be introduced to illustrate and to document the ideas.
Author Response

(The authors gave the same response as above.)
